# The Non-Specific Drp1 Inhibitor Mdivi-1 Has Modest Biochemical Antioxidant Activity

**DOI:** 10.3390/antiox11030450

**Published:** 2022-02-24

**Authors:** Evan A. Bordt, Naibo Zhang, Jaylyn Waddell, Brian M. Polster

**Affiliations:** 1Center for Shock, Trauma and Anesthesiology Research, Department of Anesthesiology, University of Maryland School of Medicine, Baltimore, MD 21201, USA; naibo.zhang@som.umaryland.edu; 2Lurie Center for Autism, Department of Pediatrics, Massachusetts General Hospital for Children, Harvard Medical School, Boston, MA 02129, USA; 3Program in Neuroscience, University of Maryland School of Medicine, Baltimore, MD 21201, USA; 4Department of Pediatrics, University of Maryland School of Medicine, Baltimore, MD 21201, USA; jwaddell@som.umaryland.edu

**Keywords:** ROS, oxidative stress, free radical, scavenger, superoxide, oxygen, mitochondria, Complex I, mitochondrial fission, neurons

## Abstract

Mitochondrial division inhibitor-1 (mdivi-1), a non-specific inhibitor of Drp1-dependent mitochondrial fission, is neuroprotective in numerous preclinical disease models. These include rodent models of Alzheimer’s disease and ischemic or traumatic brain injury. Among its Drp1-independent actions, the compound was found to suppress mitochondrial Complex I-dependent respiration but with less resultant mitochondrial reactive oxygen species (ROS) emission compared with the classical Complex I inhibitor rotenone. We employed two different methods of quantifying Trolox-equivalent antioxidant capacity (TEAC) to test the prediction that mdivi-1 can directly scavenge free radicals. Mdivi-1 exhibited moderate antioxidant activity in the 2,2′-azinobis (3-ethylbenzothiazoline 6-sulfonate) (ABTS) assay. Half-maximal ABTS radical depletion was observed at ~25 μM mdivi-1, equivalent to that achieved by ~12.5 μM Trolox. Mdivi-1 also showed antioxidant activity in the α, α-diphenyl-β-picrylhydrazyl (DPPH) assay. However, mdivi-1 exhibited a reduced capacity to deplete the DPPH radical, which has a more sterically hindered radical site compared with ABTS, with 25 μM mdivi-1 displaying only 0.8 μM Trolox equivalency. Both assays indicate that mdivi-1 possesses biochemical antioxidant activity but with modest potency relative to the vitamin E analog Trolox. Future studies are needed to evaluate whether the ability of mdivi-1 to directly scavenge free radicals contributes to its mechanisms of neuroprotection.

## 1. Introduction

Mitochondrial division inhibitor-1 (mdivi-1) crosses the blood–brain barrier and confers neuroprotection in animal models of traumatic brain injury [1,2,3,4] and focal and global ischemia [5,6,7], with protective effects in the latter comparable to that of hypothermia [7]. As mdivi-1 also mitigates chronic neurodegeneration in many additional disease models, e.g., of Alzheimer’s disease [8] and Parkinson’s disease [9,10], there is widespread interest in this compound as a preclinical drug candidate.

Mdivi-1 was originally identified as a yeast mitochondrial fission inhibitor from a chemical library screen [11]. Widely referred to as a Drp1-specific inhibitor, mdivi-1 negligibly inhibits the GTPase activity of recombinant human mitochondrial fission protein Drp1 [11,12,13,14], although it does preserve elongated mitochondrial morphology in a variety of mammalian injury models [11,15,16]. Mdivi-1 appears to attenuate mitochondrial fission, at least in some cases, by preventing the phosphorylation-dependent translocation of Drp1 to mitochondria [14,17,18]. However, although mdivi-1 may behave like a Drp1 antagonist in a variety of scenarios, many Drp1-independent mdivi-1 effects on mammalian cells have been described [13,14,19,20,21,22,23,24,25,26]. Thus, the compound is clearly not a *specific* inhibitor of Drp1. Because mdivi-1 is so broadly neuroprotective, interest in defining the potentially multiple mechanisms of action is high.

Several studies, using a variety of disease models, have shown mdivi-1 antioxidant effects, e.g., [14,18,27,28]. We previously identified mitochondrial Complex I as a Drp1-independent site of action [13]. Most Complex I inhibitors stimulate reactive oxygen species (ROS) generation [29]. However, in contrast to the classical Complex I inhibitor rotenone, mdivi-1 failed to increase ROS accumulation by primary rat cortical neurons despite impairing respiration [13]. Furthermore, compared with rotenone, mdivi-1 negligibly stimulated ROS production by isolated brain mitochondria oxidizing Complex I-linked substrates. However, like rotenone, mdivi-1 attenuated reverse electron transfer (RET)-stimulated ROS production from the quinone (Q) site of brain mitochondria [13], supporting a Complex I-dependent mechanism of action [29].

Given the divergent effects of mdivi-1 on ROS measurements relative to classical Complex I inhibitors, we wondered whether mdivi-1 has intrinsic biochemical antioxidant activity potentially contributing to the observed differences. In this study, we employed the ABTS and DPPH free radical depletion assays to test the prediction that mdivi-1 can directly scavenge free radicals. The results are consistent with this prediction. However, they indicate that mdivi-1’s antioxidant capacity is relatively modest compared with the reference antioxidant Trolox.

## 2. Materials and Methods

### 2.1. Chemicals

The following chemicals and enzymes (catalogue numbers in parentheses) were purchased from Millipore-Sigma (St. Louis, MO, USA): 6-hydroxy-2,5,7,8-tetramethylchroman-2-carboxylic acid (Trolox, #648471), 2,2′-azino-bis(3-ethylbenzothiazoline-6-sulfonic acid) diammonium salt (ABTS, #A1888), 2,2-diphenyl-1-picrylhydrazyl (DPPH, #D9132), potassium persulfate (#216224), xanthine (#X0626), L-glutathione reduced (#G4251), xanthine oxidase from bovine milk (#X1875), Cu/Zn superoxide dismutase from bovine erythrocytes (#S5395), and horseradish peroxidase (#P8375). Amplex^TM^ UltraRed Reagent (#A36006) was obtained from Thermo Fisher Scientific (Waltham, MA, USA). Other chemicals (e.g., KCl, K_2_HPO_4_, MgCl_2_, etc.) were obtained at high purity (≥99%) from Millipore-Sigma.

Mdivi-1 was purchased from Millipore-Sigma (#M0199) and prepared as a 50 mM stock in dimethyl sulfoxide (DMSO). A 1:500 dilution of DMSO (0.2%), equivalent to the highest tested mdivi-1 concentration (100 μM), was used as the vehicle control for all experiments.

### 2.2. Absorbance Spectra

To assess the absorbance spectra of mdivi-1 and DMSO vehicle, we measured optical density (O.D.) every 10 nm from 340 nm to 750 nm using a Spectramax ABS spectrophotometer (Molecular Devices, San Jose, CA, USA).

### 2.3. ABTS Radical Depletion Assay

To measure the antioxidant activity of mdivi-1, we performed an assay as described in [30] that is based on the inhibition of the absorbance of the radical cation of ABTS (ABTS^•+^). ABTS (7 mM) was prepared in water, and ABTS^•+^ (Figure 1A) was produced by combining ABTS with 2.45 mM potassium persulfate and allowing the mixture to react at room temperature in the dark for 16–24 h prior to experimental use. Trolox standard and mdivi-1 compound were mixed with ABTS^•+^, and absorbance at 734 nm was determined on a Varian Cary 100 Bio spectrophotometer (Walnut Creek, CA, USA) after a 6 min incubation. The antioxidant activity of mdivi-1 was calculated as the percentage inhibition of ABTS^•+^ solution absorbance equated against a Trolox standard curve.

### 2.4. DPPH Radical Depletion Assay

To measure the antioxidant activity of mdivi-1, we also performed an assay based on the inhibition of the absorbance of the radical DPPH^•^ (Figure 1B), as previously described [31]. Mdivi-1 and Trolox were diluted from 50 mM stocks, prepared in DMSO and water, respectively, to 2x stocks in absolute ethanol. Then, 1 mL of 100 mM DPPH made in absolute ethanol was added to 1 mL of 2x mdivi-1 or Trolox to obtain the final reaction mixtures with the indicated concentrations of compounds. The reaction mixtures were vortexed vigorously for 10 s and allowed to stand at room temperature, protected from light, for 30 min. Absorbance was determined at 517 nm on a Varian Cary 100 Bio spectrophotometer. Mdivi-1 antioxidant activity was calculated as the percentage inhibition of DPPH^•^ solution absorbance equated against a Trolox standard curve.

### 2.5. Amplex UltraRed Assay

The hydrogen peroxide (H_2_O_2_)-sensitive fluorescent indicator Amplex UltraRed (Waltham, MA, USA), an Amplex Red variant with increased sensitivity [32], was used to test for mdivi-1 H_2_O_2_-scavenging ability using an enzymatic H_2_O_2_-generating system. This system consisted of xanthine oxidase, its substrate xanthine, and superoxide dismutase [33]. Amplex UltraRed (5 μM), horseradish peroxidase (10 U/mL), superoxide dismutase 1 (SOD1, 40 U/mL), xanthine (100 μM), and test compound (DMSO, mdivi-1, Trolox, or reduced glutathione) were added to KCl assay medium consisting of 125 mM KCl, 2 mM K_2_HPO_4_, 0.1 mM EGTA, 1 mM MgCl_2_, and 20 mM HEPES, pH 7.0. Fluorescence was monitored at 37 °C using a Thermo Fisher Varioskan^TM^ Lux multimode microplate reader (excitation 560 nm; emission 585 nm). After two minutes of baseline recording to evaluate background Amplex UltraRed oxidation in the absence and presence of test compounds, xanthine oxidase (1.33 mU/mL) was added to initiate H_2_O_2_ production, and fluorescence was monitored for an additional five minutes. Amplex UltraRed oxidation rate was calculated as the slope of the first three minutes of fluorescence recordings, which was linear, as determined by linear regression analysis.

### 2.6. Data Analysis and Statistics

Statistical analyses were performed using GraphPad Prism Version 9.1 software (San Diego, CA, USA). Experimental results are expressed as means ± standard error of the mean (SEM) of 3–4 replicates. One- or two-way analysis of variance (ANOVA) followed by Bonferroni’s multiple comparisons post hoc test was used to test for differences among groups. Results with *p* < 0.05 are considered statistically significant. The [agonist] vs. response–variable slope non-linear fit model of GraphPad Prism 9.1 was used to determine the effective concentration (EC_50_) values for mdivi-1 and Trolox. The EC_50_ values give the concentration of antioxidant required to decrease the initial concentration of free radical by 50%.

## 3. Results

### 3.1. Mdivi-1 and DMSO Absorbance Spectra

Mdivi-1 (structure given in Figure 1C) was prepared in DMSO, a solvent reported to not interfere with the ABTS or DPPH radical depletion assays that rely on 734 nm and 517 nm absorbance measurements, respectively [30,34]. Because we noticed previously that the spectral properties of mdivi-1 at 340 nm interferes with NADH-based Complex I assays [13], we measured broad absorbance spectra from 340 to 750 nm to ensure that mdivi-1 absorbance did not also interfere with these antioxidant activity assays. Both mdivi-1 and its vehicle, DMSO, showed little absorbance at wavelengths ≥400 nm in solution (Figure 1D), confirming that neither interfered with the ABTS and DPPH decoloration assays.

### 3.2. Antioxidant Capacity of Mdivi-1 Measured by ABTS Radical Depletion Assay

To determine whether mdivi-1 possesses the intrinsic ability to scavenge free radicals, we first employed the ABTS radical cation assay [30]. As seen in Figure 2A, Trolox (gray squares) dose-dependently depleted ABTS^•+^ absorbance, with an EC_50_ of 12.88 μM. Although not as potent as Trolox, mdivi-1 nevertheless showed significant and dose-dependent free radical-scavenging activity over the range of concentrations typically employed to inhibit mitochondrial fission in cell-based experiments (Figure 2A; black circles). The calculated EC_50_ for mdivi-1 was 24.38 μM, approximately twice that of Trolox. Figure 2B shows the Trolox-equivalent antioxidant capacity (TEAC) of mdivi-1 for the range of concentrations used in this experiment.

### 3.3. Antioxidant Capacity of Mdivi-1 Measured by DPPH Radical Depletion Assay

To determine whether the ability of mdivi-1 to scavenge ABTS^•+^ extends to other radicals, we next tested the compound in the DPPH radical decoloration assay [31]. As seen in the ABTS^•+^ assay, Trolox efficiently depleted DPPH radical absorbance, with an EC_50_ of 11.75 μM, close to that for ABTS^•+^ depletion (12.88 μM, Figure 3A; gray squares). However, mdivi-1 exhibited a relatively weak ability to deplete the DPPH radical compared with either that of the reference antioxidant Trolox (Figure 3A; black circles) or with its own ability to scavenge ABTS^•+^ (Figure 2).

We next performed a regression analysis to directly compare the Trolox-equivalent antioxidant capacity values for mdivi-1 obtained using the ABTS radical assay with those obtained by the DPPH radical depletion assay. A significant linear correlation of mdivi-1’s TEAC was found between the ABTS and DPPH assays (Figure 4; R^2^ = 0.8961, *p* = 0.0004). However, for every mdivi-1 concentration tested, the compound depleted ABTS^•+^ to a much greater extent than the DPPH radical. Therefore, although mdivi-1 showed significant antioxidant activity in both free radical depletion assays, the antioxidant strength, calculated as Trolox equivalency, was much lower when measured by the DPPH radical decoloration assay.

Mdivi-1, like the physiological antioxidant glutathione, has a thiol group that may quench peroxides. DPPH radical is quenched by antioxidants either by electron transfer, which is very fast, or by hydrogen atom transfer, which is slow and diffusion-controlled [35,36]. Glutathione reacts with DPPH radical primarily by the latter mechanism and exhibits an almost seven-fold slower reaction rate with DPPH than the reference antioxidant Trolox, resulting in an underestimation of its antioxidant capacity when measured by the standard DPPH protocol [35].

Because mdivi-1, like glutathione, may react with peroxides, we tested whether the hydrogen peroxide (H_2_O_2_)-sensitive fluorescent indicator Amplex UltraRed (hereafter called Amplex) together with an enzymatic H_2_O_2_-generating system [33] could be used to evaluate mdivi-1’s H_2_O_2_-scavenging ability. The oxidation of the xanthine substrate by the xanthine oxidase enzyme generates superoxide, which can be rapidly converted to H_2_O_2_ by Cu/Zn superoxide dismutase (SOD1). These reactions yield an H_2_O_2_-dependent linear increase in Amplex fluorescent product catalyzed by horseradish peroxidase [33].

Of all the treatments, only glutathione increased Amplex fluorescence when incubated in the enzyme/substrate reaction mixture in the absence of xanthine oxidase. The auto-oxidation of glutathione is known to interfere with H_2_O_2_ detection by Amplex Red [37]. Once xanthine oxidase was added to trigger H_2_O_2_ production, mdivi-1, the DMSO vehicle, Trolox, and the H_2_O_2_-eliminating enzyme catalase (used as a positive control) all significantly inhibited the Amplex oxidation rate (Figure 5A,B). For the glutathione treatment, the background Amplex oxidation rate (not shown) was subtracted from the rate measured following xanthine oxidase addition. The resultant rate (shown as glutathione corrected (corr) in Figure 5B) was also significantly suppressed compared with the xanthine oxidase (XO) only control (Ctrl) condition (solid black trace in Figure 5A). Though the mdivi-1 solutions, like glutathione, significantly inhibited Amplex oxidation relative to Ctrl, the level of inhibition was not dose dependent across the tested mdivi-1 concentration range (25–100 μM, dashed lines in Figure 5A). In addition, the extent of inhibition did not differ from the DMSO vehicle control. Thus, the significant effect of DMSO confounded the measurement of the peroxide-scavenging ability of mdivi-1 using this assay.

We also attempted to evaluate whether mdivi-1 can scavenge superoxide using a combination of xanthine, xanthine oxidase, and the fluorescent indicator dihydroethidium. However, in our hands, this method lacked sufficient sensitivity for accurate rate determination in cell-free assays (data not shown).

## 4. Discussion

Mdivi-1 is a promising preclinical drug candidate for both acute and chronic neurodegenerative disorders. The compound is referred to as a specific Drp1 inhibitor in many, if not most, of the ~430 peer reviewed journal articles in which it has been mentioned to date (see [15,38] for review). However, several biological effects of mdivi-1 are now described in Drp1-deficient cells, indicating Drp1-independent mechanisms of action [38]. These include antioxidant effects [14].

The very specific goal of the experiments described in this short report was to determine whether mdivi-1 can directly scavenge free radicals, an ability that could potentially contribute to its Drp1-independent biological activities in cells. The findings obtained by two of the most widely used assays to evaluate antioxidant strength demonstrate that mdivi-1 can scavenge free radicals in cell-free systems. We found a linear relationship across a range of mdivi-1 concentrations in the TEAC values obtained by the ABTS and DPPH radical depletion assays. However, a limitation of our study is that mdivi-1 far more effectively scavenges ABTS radicals than DPPH radicals, hindering the interpretation of its antioxidant capacity.

Though still one of the most widely used assays for evaluating antioxidant activity, a re-evaluation of the DPPH^•^-based method found that despite measuring free radical-scavenging ability, it is a poor method for ranking antioxidant strength [35]. The main reasons for this are that TEAC values determined by the DPPH assay are influenced by whether a compound has bulky ring adducts and/or multiple phenolic rings that limit accessibility to the sterically hindered DPPH radical site (Figure 1B) and are additionally influenced by the nature of the compound’s dominant antioxidant mechanism (fast vs. slow) [39]. The DPPH method, relative to the ABTS method, is also more sensitive to pH effects [39]. Vanillic acid and resorcinol, in addition to the aforementioned glutathione, are examples of antioxidants that exhibit low antioxidant capacity relative to Trolox in the DPPH assay [35]. Various other compounds were found to have weaker antioxidant capacity when measured by the DPPH radical depletion assay as compared to the ABTS radical depletion assay [40,41,42,43].

Another limitation of our study is that neither ABTS^•+^ nor DPPH^•^ are free radicals found in mammals. Mdivi-1 has a thiol group (see Figure 1C) that most likely quenches peroxides and sulphenic oxidations. Mdivi-1′s ability to scavenge physiological ROS may be superior or inferior to its ability to scavenge ABTS^•+^ or DPPH^•^. With this caveat in mind, we explored the use of an enzymatic system that generates H_2_O_2_ combined with detection by Amplex UltraRed, an assay occasionally used to evaluate antioxidant activity [44,45]. Unfortunately, the significant effect of the DMSO solvent obscured the interpretation of the results. DMSO has known antioxidant properties, including the ability to scavenge hydroxyl radicals [46] and inhibit lipid peroxidation [47]. A drawback of the Amplex (Ultra) Red assay is that the antioxidant test compound may interact with one or more of the enzymes (e.g., xanthine oxidase, horseradish peroxidase) or the fluorescent indicator, making results difficult to interpret (see [37,44,45]). For this reason, methods that directly monitor radical disappearance, such as the ABTS and DPPH decolorization assays, are generally preferred. Because we saw a significant DMSO vehicle effect, we did not attempt to determine whether the effects observed in the Amplex assay were due to H_2_O_2_ scavenging or the inhibition of enzyme activity.

Both antioxidant and pro-oxidant effects of mdivi-1 have been reported in vitro and in vivo [15,38]. These opposing mdivi-1 actions may result from the conditions under which it interacts with Complex I or other protein targets rather than from direct free radical interactions. We found previously that although mdivi-1 fails to increase ROS in primary cortical neurons, it stimulates ROS to the same extent as the Complex I inhibitor rotenone in both wild-type and Drp1 knockout mouse embryonic fibroblasts [13]. Future work is needed to determine whether mdivi-1 binds to the Complex I Q-site, like rotenone, or exhibits an alternative mechanism of Complex I inhibition. Interestingly, idebenone, widely considered an antioxidant, is a compound that, like mdivi-1, both inhibits Complex I activity and exhibits relatively weak antioxidant activity in cell-free assays [48,49]. Recent work found that the expression level of NAD(P)H quinone dehydrogenase 1 (NQO1), an enzyme that can reduce idebenone to idebenol, dictates whether the compound behaves as an antioxidant or pro-oxidant in cells [50,51]. Future elucidation of the precise Drp1-independent targets of mdivi-1 may similarly identify enzymes that modify its redox behavior in cells.

## 5. Conclusions

In summary, using two of the most widely employed free radical depletion assays for evaluating antioxidant capacity, our study demonstrated that mdivi-1 can directly scavenge free radicals. Mdivi-1 exhibited significant antioxidant activity in both assays but showed a superior ability to scavenge ABTS^•+^ compared with the DPPH^•^ radical that has a sterically hindered reaction site. Given that the antioxidant potency of mdivi-1 is ~two-fold lower than the vitamin E analog Trolox, even as determined by the ABTS^•+^ depletion assay, it seems unlikely that biochemical antioxidant activity is the major mechanism responsible for its neuroprotection. Nevertheless, the ability of mdivi-1 to engage in redox reactions independently of actions on its putative target Drp1 is another variable to consider when attributing biological effects to specific molecular mechanisms.

## Figures and Tables

**Figure 1 antioxidants-11-00450-f001:**
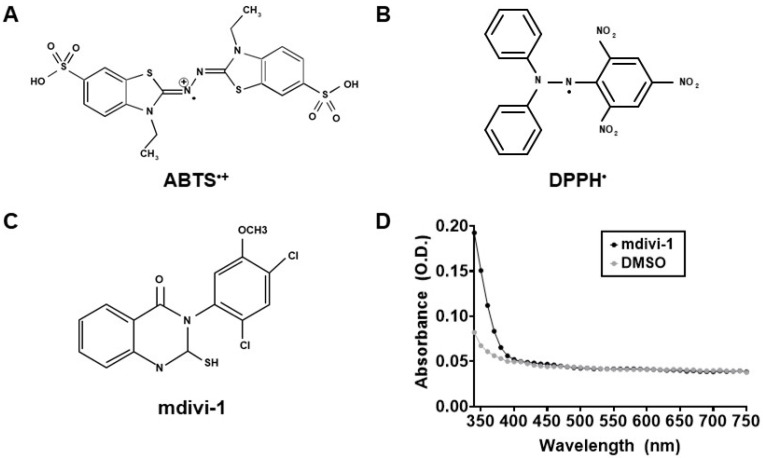
Mitochondrial division inhibitor-1 (mdivi-1) structure, absorbance, and the free radicals used to measure its antioxidant activity. (**A**) 2,2′-Azino-bis(3-ethylbenzothiazoline-6-sulfonic acid) (ABTS) radical. (**B**) 2,2-Diphenyl-1-picrylhydrazyl (DPPH) radical. (**C**) Mdivi-1 structure. (**D**) Absorbance spectra of mdivi-1 (100 μM) and dimethyl sulfoxide (DMSO, 0.2%) in solution. O.D., optical density.

**Figure 2 antioxidants-11-00450-f002:**
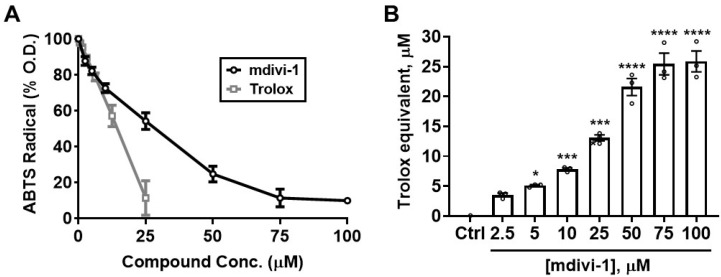
Mdivi-1 dose-dependently depletes ABTS^•+^. (**A**) Mdivi-1 (2.5–100 μM) or Trolox (0.78–25 μM) was added to ABTS^•+^ and antioxidant activity was monitored by the ABTS^•+^ decolorization assay as a decrease in optical density (O.D.), expressed as a percentage of the initial O.D. Each value is depicted as the mean ± SEM; *n* = 3 separate experiments. (**B**) Trolox-equivalent antioxidant capacity (TEAC) of mdivi-1 for the depletion of ABTS^•+^. Bar graphs represent mean ± SEM from *n* = 3 separate experiments. One-way ANOVA, F-value (7, 16) = 95.29, *p* < 0.0001, was followed by Bonferroni’s multiple comparisons post hoc test comparing mdivi-1 treatment to the vehicle control (Ctrl). * *p* < 0.05, *** *p* < 0.001, **** *p* < 0.0001.

**Figure 3 antioxidants-11-00450-f003:**
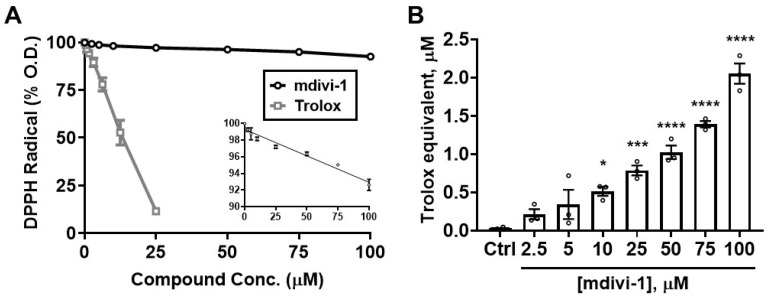
Mdivi-1 weakly but dose-dependently depletes DPPH radical. (**A**) Mdivi-1 (2.5–100 μM) or Trolox (0.78–25 μM) was added to DPPH radical and antioxidant activity was monitored by the DPPH decolorization assay as a decrease in optical density (O.D.), expressed as a percentage of the initial O.D. Each value is depicted as the mean ± SEM from *n* = 3 separate experiments. The inset in (**A**) shows the means ± SEM for the mdivi-1 group on a compressed scale (DPPH Radical (% O.D.), y-axis vs. Compound Conc. (μM), x-axis), fit by linear regression analysis (R^2^ = 0.9139, *p* < 0.001). (**B**) Trolox-equivalent antioxidant capacity (TEAC) of mdivi-1 for the depletion of DPPH. Bar graphs represent means ± SEM from *n* = 3 separate experiments. One-way ANOVA, F-value (7, 16) = 48.44, *p* < 0.0001, was followed by Bonferroni’s multiple comparisons post hoc test comparing mdivi-1 treatment to Ctrl. * *p* < 0.05, *** *p* < 0.001, **** *p* < 0.0001.

**Figure 4 antioxidants-11-00450-f004:**
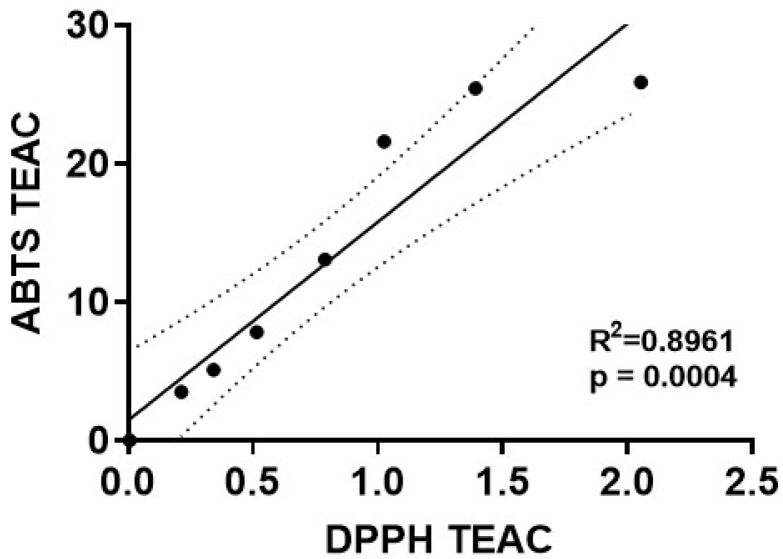
Correlation of mdivi-1’s dose-dependent antioxidant capacity values measured by two different radical depletion assays. The Trolox-equivalent antioxidant capacity (TEAC) of mdivi-1 in the ABTS^•+^ (y-axis) and DPPH radical (x-axis) depletion assays was found to be significantly correlated by linear regression analysis (R^2^ = 0.8961, *p* = 0.0004). Dotted line depicts 95% confidence bands for the best-fit line. Note that the axes have different scales.

**Figure 5 antioxidants-11-00450-f005:**
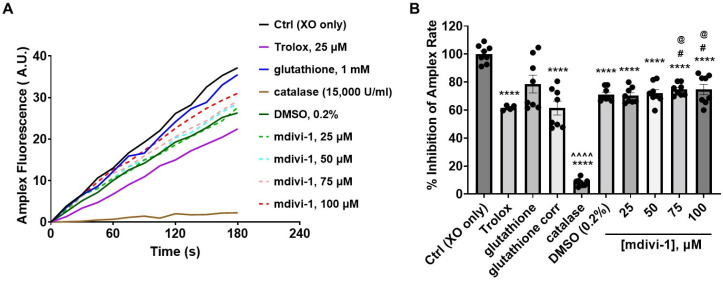
Effect of mdivi-1 on Amplex UltraRed oxidation rate. (**A**) Fluorescence in arbitrary units (A.U.) of Amplex UltraRed (Amplex) oxidation product monitored over time following xanthine oxidase (XO) addition in the absence (Ctrl, control) or presence of mdivi-1 (25–100 μM), DMSO (0.2%, vehicle for mdivi-1), Trolox (25 μM), glutathione (1 mM) or catalase (15,000 U/mL). Glutathione was prepared from powder in the aqueous Amplex assay medium and Trolox was diluted from a 50 mM stock prepared in water. Traces are means of quadruplicate wells and are representative of the experiment run on two different days. Trolox was only included on one of the days (depicted). (**B**) Amplex UltraRed oxidation rates expressed as a percentage of the Ctrl rate. Data are mean ± SEM, *n* = 4 for Trolox, and *n* = 8 for all others. An uncorrected representative glutathione trace is shown in (**A**), and both the uncorrected and corrected (corr, i.e., background-subtracted) rates are given in (**B**). Only the corrected rates were used for statistical analysis. A two-way mixed-model ANOVA followed by Bonferroni’s multiple comparisons post hoc test was used to test for differences among treatments, with treatment and experiment day as factors. Both treatment (F-value (8, 58) = 97.25, *p* < 0.0001) and the experiment day (F-value (1, 58) = 10.50, *p* = 0.002) had a significant effect, accounting for 92.18% and 1.244% of the total variation, respectively. **** *p* < 0.0001 compared to Ctrl, # *p* < 0.05 compared to Trolox, @ *p* < 0.05 compared to glutathione corr, ^^^^ *p* < 0.0001 compared to all other groups.

## Data Availability

Data is contained within the article.

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
