# Peer review of "The Non-Specific Drp1 Inhibitor Mdivi-1 Has Modest Biochemical Antioxidant Activity"

_antioxidants, 2022, doi:10.3390/antiox11030450_

Round 1

Reviewer 1 Report

Tough the authors have not been able to use the DHE assay, they have adressed and discuss all my concerns properly.

Author Response

We thank the reviewer for the effort in re-reviewing the manuscript and noting that concerns have been properly addressed.

Reviewer 2 Report

The authors have provided a new figure to the manuscript since the last round of revision, however this has raised concerns regarding the proper use of solvent for solubilizing the mdivi-1 inhibitor (see the major points below).

Major points:

1) There seems to be the following issues with the data presented in Figure 5:

a) Although the authors used ethanol to dissolve mdivi-1 according to their claim "One mL of 100 mM DPPH made in absolute ethanol was added to 1 mL of mdivi-1 or Trolox in absolute ethanol" (line 105), they have used DMSO as mdivi-1 solvent for the Amplex UltraRed assay. Would replacing DMSO with ethanol help to resolve the solvent effect observed in Figure 5?

b) The authors claim that "mdivi-1 ... significantly inhibited Amplex oxidation" in "Though mdivi-1, like glutathione, significantly inhibited Amplex oxidation relative to Ctrl, the level of inhibition was not dose-dependent across the tested mdivi-1 concentration range (25 – 100 uM, dashed lines in Figure 5A)" (line 195), despite the observed solvent effect of DMSO.

c) It is not clear what solvent was used and what was its dilution for Trolox in Figure 5?

d) Lastly, the Amplex UltraRed assay might become more sensitive upon increasing xanthine oxidase activity.

2) Do different solvents impact on the ability of mdivi-1 to inhibit mitochondrial fission in live cells?

Minor points:

1) Please change "2,2’-Azino-bis(3-ethylbenzothiazoline-6-sulfonic acid)" to "2,2’-azino-bis(3-ethylbenzothiazoline-6-sulfonic acid)" (line 75).

2) Please replace "2,2-Diphenyl-1-picrylhydrazyl" with "2,2-diphenyl-1-picrylhydrazyl" (line 76).

3) Please change "ThermoFisher" to "Thermo Fisher" (lines 79, 121).

4) Please replace "radical cation of ABTS" with "radical cation of ABTS (ABTS•+)" (line 94) and "ABTS radical cation (ABTS•+, Figure 1A)" with "ABTS•+ (Figure 1A)" (line 95).

5) Please replace "16-24" with "16–24" (line 96).

6) Please change "37°C" to "37 °C" (line 121).

7) Please replace "3-4" with "3–4" (line 131).

8) It is not clear what the authors mean by "four parameters" in "Results with p < 0.05 are considered statistically significant. The [Agonist] vs. response – Variable slope (four parameters) non-linear fit model of GraphPad Prism 9.1 was used to determine the effective concentration (EC50) values for mdivi-1 and Trolox" (line 133)?

9) Please change "[Agonist] vs. response – Variable" to "[agonist] vs. response – variable" (line 134).

10) Please replace "350-750" with "350–750" (line 145).

11) Please replace "ABTS radical" with "ABTS•+" (lines 152, 159, 169, 266, legend to Figures 2 and 4).

12) Please define "F" (legend to Figure 2, 3, 5).

13) Please change "(12.88 uM) for ABTS•+ depletion" to "for ABTS•+ depletion (12.88 uM)" (line 162).

14) Please replace "SEM. n = 3" with "SEM from n = 3" (legend to Figure 3).

15) Please replace "R2= 0.9139" with "R2 = 0.9139" (legend to Figure 3).

16) Please change "R2= 0.8961" to "R2 = 0.8961" (legend to Figure 4).

17) Please replace "n=8" with "n = 8" (legend to Figure 4).

18) Please change "@p<0.05" to "@ p < 0.05" (legend to Figure 5).

19) Please replace "not" with "data not" (line 203).

20) Please change "ABTS•+ radical" to "ABTS•+" (line 264, legend to Figure 2 2x).

Author Response

The authors have provided a new figure to the manuscript since the last round of revision, however this has raised concerns regarding the proper use of solvent for solubilizing the mdivi-1 inhibitor (see the major points below).

RESPONSE: We thank the reviewer for the effort in re-reviewing the manuscript and address all concerns in point-by-point fashion below.

Major points:

1) There seems to be the following issues with the data presented in Figure 5:

  1. a) Although the authors used ethanol to dissolve mdivi-1 according to their claim "One mL of 100 mM DPPH made in absolute ethanol was added to 1 mL of mdivi-1 or Trolox in absolute ethanol" (line 105), they have used DMSO as mdivi-1 solvent for the Amplex UltraRed assay. Would replacing DMSO with ethanol help to resolve the solvent effect observed in Figure 5?

RESPONSE: We apologize for the confusion caused by imprecise description. Mdivi-1 is not soluble in ethanol at the concentrations used in this study. This description reflects mdivi-1 diluted from the 50 mM DMSO stock to twice the final tested concentration in absolute ethanol. We added the following information to the methods to clarify (page 3, lines 105-109):

“Mdivi-1 and Trolox were diluted from 50 mM stocks, prepared in DMSO and water, respectively, to 2x stocks in absolute ethanol. One mL of 100 mM DPPH made in absolute ethanol was then added to 1 mL of 2x mdivi-1 or Trolox to obtain the final reaction mixtures with the indicated concentrations of compounds.”

  1. b) The authors claim that "mdivi-1 ... significantly inhibited Amplex oxidation" in "Though mdivi-1, like glutathione, significantly inhibited Amplex oxidation relative to Ctrl, the level of inhibition was not dose-dependent across the tested mdivi-1 concentration range (25 – 100 uM, dashed lines in Figure 5A)" (line 195), despite the observed solvent effect of DMSO.

RESPONSE: We thank the reviewer for noting this imprecise phrasing. We have amended the text to read “Though the mdivi-1 solutions, like glutathione…” to clarify that the significant effects were due to the mdivi-1 solutions (i.e. containing DMSO) and cannot be attributed to the compound itself.

  1. c) It is not clear what solvent was used and what was its dilution for Trolox in Figure 5?

RESPONSE: Trolox was diluted from a 50 mM stock prepared in water. This information is now provided in the Figure 5 legend. We also now specify in the same legend that “glutathione was prepared from powder in the aqueous Amplex assay medium.”

  1. d) Lastly, the Amplex UltraRed assay might become more sensitive upon increasing xanthine oxidase activity.

RESPONSE: The DMSO vehicle effect in the assay discouraged us from pursuing this method further since mdivi-1 is only soluble in organic solvent.

2) Do different solvents impact on the ability of mdivi-1 to inhibit mitochondrial fission in live cells?

RESPONSE: Mdivi-1 is only soluble in organic solvents and, to the best of our knowledge, DMSO is the recommended solvent for mdivi-1 by all manufacturers. We are unaware of studies that have used other solvents for investigating the ability of mdivi-1 to inhibit mitochondrial fission in live cells.

Minor points:

1) Please change "2,2’-Azino-bis(3-ethylbenzothiazoline-6-sulfonic acid)" to "2,2’-azino-bis(3-ethylbenzothiazoline-6-sulfonic acid)" (line 75).

RESPONSE: Done.

2) Please replace "2,2-Diphenyl-1-picrylhydrazyl" with "2,2-diphenyl-1-picrylhydrazyl" (line 76).

RESPONSE: Done.

3) Please change "ThermoFisher" to "Thermo Fisher" (lines 79, 121).

RESPONSE: Done.

4) Please replace "radical cation of ABTS" with "radical cation of ABTS (ABTS•+)" (line 94) and "ABTS radical cation (ABTS•+, Figure 1A)" with "ABTS•+ (Figure 1A)" (line 95).

RESPONSE: Done.

5) Please replace "16-24" with "16–24" (line 96).

RESPONSE: Done.

6) Please change "37°C" to "37 °C" (line 121).

RESPONSE: Done.

7) Please replace "3-4" with "3–4" (line 131).

RESPONSE: Done.

8) It is not clear what the authors mean by "four parameters" in "Results with p < 0.05 are considered statistically significant. The [Agonist] vs. response – Variable slope (four parameters) non-linear fit model of GraphPad Prism 9.1 was used to determine the effective concentration (EC50) values for mdivi-1 and Trolox" (line 133)?

RESPONSE: “Four parameters” is an alternative name for the variable slope model. The GraphPad Prism website (https://www.graphpad.com/guides/prism/latest/curve-fitting/reg_dr_stim_variable_2.htm) includes the following description “Many dose-response curves have a standard slope of 1.0. This model does not assume a standard slope but rather fits the Hill Slope from the data, and so is called a Variable slope model. This is preferable when you have plenty of data points. It is also called a four-parameter dose-response curve, or four-parameter logistic curve, abbreviated 4PL.” To avoid confusion, we deleted “(four parameters).”

9) Please change "[Agonist] vs. response – Variable" to "[agonist] vs. response – variable" (line 134).

RESPONSE: Done.

10) Please replace "350-750" with "350–750" (line 145).

RESPONSE: Done.

11) Please replace "ABTS radical" with "ABTS•+" (lines 152, 159, 169, 266, legend to Figures 2 and 4).

RESPONSE: Done.

12) Please define "F" (legend to Figure 2, 3, 5).

RESPONSE: We have changed “F” to “F-value.” The F-value is a standard value calculated when an ANOVA is run. It equals the variation between sample means / variation within the samples.

13) Please change "(12.88 uM) for ABTS•+ depletion" to "for ABTS•+ depletion (12.88 uM)" (line 162).

RESPONSE: Done.

14) Please replace "SEM. n = 3" with "SEM from n = 3" (legend to Figure 3).

RESPONSE: Done.

15) Please replace "R2= 0.9139" with "R2 = 0.9139" (legend to Figure 3).

RESPONSE: Done.

16) Please change "R2= 0.8961" to "R2 = 0.8961" (legend to Figure 4).

RESPONSE: Done.

17) Please replace "n=8" with "n = 8" (legend to Figure 4).

RESPONSE: Done.

18) Please change "@p<0.05" to "@ p < 0.05" (legend to Figure 5).

RESPONSE: Done.

19) Please replace "not" with "data not" (line 203).

RESPONSE: Done.

20) Please change "ABTS•+ radical" to "ABTS•+" (line 264, legend to Figure 2 2x).

RESPONSE: Done.

Round 2

Reviewer 2 Report

The authors have successfully addressed all previous concerns.